# Unusual Association of Diamond–Blackfan Anemia and Severe Sinus Bradycardia in a Six-Month-Old White Infant: A Case Report and Literature Review

**DOI:** 10.3390/medicina59020362

**Published:** 2023-02-14

**Authors:** Stefana Maria Moisa, Elena-Lia Spoiala, Laura Mihaela Trandafir, Lacramioara Ionela Butnariu, Ingrith-Crenguta Miron, Antonela Ciobanu, Adriana Mocanu, Anca Ivanov, Carmen Iulia Ciongradi, Ioan Sarbu, Anamaria Ciubara, Carmen Daniela Rusu, Alina Costina Luca, Alexandru Burlacu

**Affiliations:** 1Pediatrics Department, Faculty of Medicine, “Grigore T. Popa” University of Medicine and Pharmacy, 700115 Iasi, Romania; 2“Sfanta Maria” Clinical Emergency Hospital for Children, 700309 Iasi, Romania; 3Genetics Department, Faculty of Medicine, “Grigore T. Popa” University of Medicine and Pharmacy, 700115 Iasi, Romania; 42nd Department of Surgery—Pediatric Surgery and Orthopedics, “Grigore T. Popa” University of Medicine and Pharmacy, 700115 Iasi, Romania; 5Faculty of Medicine, “Dunarea de Jos” University of Medicine and Pharmacy, 800008 Galati, Romania; 6Faculty of Medicine, “Grigore T. Popa” University of Medicine and Pharmacy, 700115 Iasi, Romania; 7Institute of Cardiovascular Diseases “Prof. Dr. George I.M. Georgescu”, 700503 Iasi, Romania

**Keywords:** Diamond–Blackfan anemia, severe persistent sinus bradycardia, rare diseases association

## Abstract

Diamond–Blackfan anemia is a rare (6–7 million live births), inherited condition manifesting as severe anemia due to the impaired bone marrow production of red blood cells. We present the unusual case of a six month old infant with a de novo mutation of the *RPS19* gene causing Diamond–Blackfan anemia who additionally suffers from severe sinus bradycardia. The infant was diagnosed with this condition at the age of four months; at the age of 6 months, she presents with severe anemia causing hypoxia which, in turn, caused severe dyspnea and polypnea, which had mixed causes (hypoxic and infectious) as the child was febrile. After correction of the overlapping diarrhea, metabolic acidosis, and severe anemia (hemoglobin < 3 g/dL), she developed severe persistent sinus bradycardia immediately after mild sedation (before central venous catheter insertion), not attributable to any of the more frequent causes, with a heart rate as low as 49 beats/min on 24 h Holter monitoring, less than the first percentile for age, but with a regular QT interval and no arrhythmia. The echocardiogram was unremarkable, showing a small interatrial communication (patent foramen ovale with left-to-right shunting), mild left ventricular hypertrophy, normal systolic and diastolic function, and mild tricuspid regurgitation. After red cell transfusion and appropriate antibiotic and supportive treatment, the child’s general condition improved dramatically but the sinus bradycardia persisted. We consider this a case of well-tolerated sinus bradycardia and foresee a good cardiologic prognosis, while the hematologic prognosis remains determined by future corticoid response, treatment-related complications and risk of leukemia.

## 1. Introduction

Diamond–Blackfan anemia (DBA) is a rare inherited bone marrow failure syndrome [1]. Most of the cases are diagnosed in infancy or early childhood, with a median age at a presentation of 2 months old [2]. DBA is the result of haploinsufficiency for ribosomal protein (RP) genes, including *RPS19*, *RPS24*, *RPS17*, *RPL35A*, *RPL5*, *RPL11*, *RPS7*, *RPS10*, *RPS26*, *RPL26*, *RPL15*, *RPL31*, *RPS29*, *RPS28*, *RPL27*, *RPS27*, *RPS15A*, *RPL35*, *RPL18*, *GATA1* and *TSR2* [3]. Neonatal anemia and occasional dysmorphism are the main features of the classical presentation [1]. Approximately 80% of patients respond to corticotherapy, and 30% result in transfusion and/or corticosteroid dependency [4].

HLA-matched sibling stem cell transplantation has favorable outcomes [5,6,7]. Mortality is mainly derived from treatment complications (iron toxicity, infections, complications related to stem cell transplant) but also from neoplastic conditions, which are more frequently reported in DBA than in the general population (solid tumors, leukemia, myelodysplastic syndrome) [8,9,10]. DBA is characterized by a broad spectrum of phenotypes. Almost 50% of cases present with physical abnormalities. Among these, the most common congenital physical abnormalities include thumb and upper extremity malformations, craniofacial anomalies, and short stature [2]. Several congenital heart defects were reported in the DBA Registry (DBAR): ventricular septal defect (44%), atrial septal defect (32.5%), coarctation of the aorta (4.4%), pulmonic valve stenosis (5.3%), tetralogy of Fallot (3.5%), aortic valve stenosis (2.6%), and total anomalous pulmonary venous return (0.9%) [11]. Herein, we report an unusual association of DBA with severe sinus bradycardia in a six-month-old white infant.

## 2. Case Report

We present the case of a six month old patient who was first diagnosed with DBA type 1 (OMIM #105650) at the age of four months when she was admitted to the Hematology ward of our hospital. The diagnosis had been made using a gene panel by Invitae laboratories (USA) (Invitae Bone Marrow Failure Syndromes Panel) that identified the pathogenic variant NM_001022.4:c.3G>T, p.Met1? in the gene RPS19 (located on chromosome 19q13.2) that is associated with DBA type 1 (DBA1). Genetic testing consisted of full-gene sequencing and deletion/duplication analysis using next-generation sequencing technology (NGS) (Illumina technology) for the 116 genes included in the analyzed gene panel.

The current presentation was motivated by a fever (38 degrees Celsius) that started three days before being admitted and was accompanied by feeding refusal and accentuated paleness. Upon admission, the clinical status of the infant was severely compromised: severe tachypnea (80 breaths per minute), dyspnea, rhythmic tachycardia (180 beats/min), agitation, and peripheral oxygen saturation of 84%. The patient was initially tachycardic, but she became bradycardic immediately after mild sedation for central line insertion (before central venous catheter (CVC) insertion). The patient presents craniofacial dysmorphic features with downslanting palpebral fissures, frontal bossing and micrognathia, and muscular hypotonia (Figure 1).

Initial biological tests revealed severe anemia (hemoglobin (Hb) level of less than 3 g/dL) and severe decompensated metabolic acidosis (pH = 7.11). The infant developed diarrhea during the following days, which necessitated symptomatic and antibiotic therapy (cefotaxime).

After correcting all metabolic imbalances, blood mass transfusion, and the resolution of the diarrhea, the continuous heart monitorization revealed severe sinus bradycardia (48–50 beats per minute). Concerning different time points, the following heart rates were recorded: 80 min before sedation (systolic blood pressure 80–90 mmHg), 47 min before CVC insertion and 50–80 min after CVC insertion and in the days afterwards (systolic blood pressure 85–95 mmHg). A central venous catheter FR 18 was inserted (Seldinger technique) using the following sedation strategy: ketamine 15 mg (2.5 mg/kg), midazolam 0.75 mg (0.125 mg/kg), and atropine 0.05 mg. The child was transferred to the Cardiology department for further testing.

The cardiological exam revealed bradycardic heart sounds, a systolic murmur on the upper left sternal border, normal blood pressure (90/55 mm Hg), and no peripheral edema. The child was not bradycardic when she first presented to the hospital (current and four-month-old presentation). The ECG showed sinus bradycardia 55 beats/min with normal QRS axis.

Echocardiography revealed mild left ventricular hypertrophy (5 mm), a minimum interventricular shunt (2 mm), second-degree tricuspid regurgitation, no left ventricular dilatation (septal–mitral distance = 3 mm), no pulmonary hypertension, normal pericardial fluid, and normal systolic and diastolic function.

The ECG Holter 24 h monitoring showed a minimum heart rate of 49 beats/min and a maximum heart rate of 161 beats/min (Figure 2A,B). There were no ventricular pauses longer than 2000 ms and no ventricular or supraventricular arrhythmias.

An electrophysiology specialist recommended watchful waiting (ECG to be performed once every six months, ECG Holter every year).

Since the patient later developed bradycardia, a gene panel for cardiac arrhythmias (Invitae Arrhythmia and Cardiomyopathy Comprehensive Panel, including 168 genes) was recommended, which revealed the presence of a heterozygous mutation of the DSG2 gene (c.308T>C) (p.Val103Ala), a variant of unknown significance (VUS) [12]. Whole Exome sequencing, although recommended, could not be performed as the family could not cover the cost. The *DSG2* gene is associated with autosomal dominant arrhythmogenic right ventricular cardiomyopathy (ARVC) (MedGen UID: 347543) [13].

Furthermore, the *DSG2* gene has preliminary evidence supporting a correlation with dilated cardiomyopathy (DCM) (MedGen UID: 414552) [14] and autosomal recessive arrhythmogenic right ventricular cardiomyopathy (ARVC) (PMID: 28818065, 23381804). Not all variants present in a gene cause disease. The clinical significance of the variant(s) identified in this gene is uncertain.

The patient was discharged bradycardic, but the mother was instructed on heart rate measurement methods and was advised to come to the hospital if the heart rate dropped below 50 beats/min or if symptoms of hemodynamic instability appeared. At discharge, the patient was not transfusion-dependent, and corticosteroid therapy—prednisone 2 mg/kg/day, in association with a proton pump inhibitor for three weeks—was recommended. Afterwards, the hematologic response as well as the requirement of long-term corticosteroids or transfusions will be evaluated. The patient will receive lifelong follow-up, as many patients can relapse after several years of development and might became transfusion-dependent, with a worse prognosis due to iron overload (including end-stage cardiac disease) [15].

## 3. Methods

We performed an electronic search in PubMed and Embase databases using the following keywords [“Diamond–Blackfan anemia”] AND [“children” OR “pediatric” OR “paediatric”] from the time of their inception to September 2022. We focused on both phenotypic and genotypic features. We have included only the studies in English or French. Our research was also limited to human studies.

## 4. Results

The process of the study selection is shown as a PRISMA flow diagram in Figure 3. Our electronic search retrieved a total of 786 records (366 in the PubMed database and 420 in Embase). After removing the duplicates, 432 papers were further screened by title and abstract analysis. Afterward, we excluded 365 papers due to the following reasons: studies depicting only the genetic profile without correlations to clinical presentation (167), studies focusing only on treatment complications (98), animal studies (69), Shwachman Diamond anemia cases (15), conference abstracts (10), or letters to the editor (6). The remaining 67 studies were further analyzed for eligibility by full-text reading. Finally, we have excluded the papers focusing exclusively on malignancy risk (18) or follow-up or treatment results (24), as well as papers lacking information on detailed percentages of physical anomalies (11) and papers focusing on an experimental treatment for DBA (4). In the end, we have analyzed 10 papers that provide insight into DBA cases, both from a clinical and genetic perspective (Table 1).

## 5. Discussion

We present an unusual association of two rare pathologies: severe persistent sinus bradycardia of uncertain origin and DBA. The other family members (parents and the child’s siblings) tested negative for this condition; therefore, the patient has a de novo mutation. The parents’ risk of having another affected offspring is 1–2% (because one cannot exclude germinal mosaicism in one of the parents). At adult age, the patient’s risk of having affected offspring would be 50% (if her partner were healthy), as the disease has an autosomal dominant transmission.

In our case, the severe bradycardia did not have an apparent cardiac cause. Hyper- and hypokalemia, malnutrition, hypothyroidism and hypothermia were ruled out. Additionally, hypoxia was not the cause of bradycardia in our patient, as it occurred after red cell transfusion. Transfontanelar ultrasound excluded increased intra-cranial pressure. Cardiac malformations such as the atrial septal defect could cause atrioventricular block, but not sinus bradycardia. Kawasaki disease, cardiomyopathies, myocarditis, endocarditis, rheumatic fever, and myocardial ischemia were also ruled out. No heart rate decreasing drugs were administered. The bradycardia appeared after mild sedation for central venous line insertion but did not resolve once the sedative was eliminated from the body. Since the child was hemodynamically stable and asymptomatic once the anemia and diarrhea were treated, we considered this a case of benign sinus bradycardia and expected a good prognosis, even though her heart rate values were less than the first percentile for age. Nevertheless, anemia prognosis is yet to be determined, as it is impacted by the corticoid response which will be evaluated during follow-up.

One study, involving a large cohort of pediatric patients with CVC (n = 3180), reported a low incidence of all types of cardiac arrhythmias (1% of cases): atrial arrhythmias (71%), ventricular arrhythmias (13%) and undetermined (16%) [26]. Thus, the incidence of bradycardia following CVC insertion is expected to be much lower than 1%. Additionally, there were only limited data published on bradycardia following CVC insertion, mainly case reports in adult patients [27,28]. Therefore, establishing a direct correlation between CVC insertion and bradyarrhythmias in pediatric patients might be misleading due to the lack of data in this subset of patients. Moreover, the bradycardia in our case occurred before CVC insertion.

In order to avoid cardiac arrhythmias’ occurrence due to CVC insertion, the optimal depth of CVC (superior vena cava and right junction) could be predicted by the height of the children [29]. In one study, 257 children aged <13 years were enrolled. The authors observed that the optimal CVC depth was linked to the height of the children (a stronger correlation than in the case of weight and age). The following formulas were developed for left-sided CVC insertion: 0.07 × height (for internal jugular vein) and 0.08 × height (for subclavian vein catheter). To the obtained values, the length between CVC insertion point and clavicle must be added. These formulas could accurately predict the optimal CVC depth in 98.5% and 94.0% of cases, for internal jugular vein and subclavian vein catheters, respectively [29]. Therefore, a simple approach based on patients’ height could be used to predict a correct CVC position in more than 90% of cases. Moreover, arrhythmic complications due to mechanical stimulation during CVC insertion might be avoided [29].

A height-based approach is also available for right-sided CVC insertion to predict the optimal depth. In one study, 60 right internal jugular vein CVCs were inserted [30]. The authors confirmed that the correct CVC position is highly correlated with patients’ height. They developed a formula that could predict an optimal CVC depth in 97.5% of cases: 1.7 + (0.07 × height), in case of children with 40–140 cm height [30]. Thus, arrhythmic complications could be avoided in the most of patients, by an accurate prediction of CVC insertion above the right atrium.

Taking into account that the neonatal screening for congenital hypothyroidism was normal, until new information appears in the specialized literature related to *DSG2* gene mutations involved in the occurrence of heart rhythm disorders we cannot state with certainty that the gene mutation detected in our patient’s case is correlated with the appearance of bradycardia.

Data regarding cardiac manifestation in DPA pediatric patients are scarce, addressing, mainly, accompanying congenital heart diseases (CHD) and cardiac complications due to iron overload (in transfusion-dependent DBA children) [11,31]. Compared to the general population, children with DBA exhibited a higher prevalence of CHD (atrial and ventricular septal defects, aortic coarctation or multiple defects), as was reported in the DBA registry (14.9% vs. <1%, *p* < 0.0001). Moreover, from 102 CHD patients, one was confirmed with DBA due to RPS24 mutation [11,32]. In a cohort of 31 transfusion-dependent patients with DBA, more than a half of the patients (57%) had severe iron overload [33]. One study reported that the increased production of plasma non-transferrin-bounding iron in DBA patients might explain their greater risk of iron deposits in extra-hepatic tissues (including the heart) [34]. End-stage heart failure due to iron overload cardiomyopathy (end-stage cardiac hemochromatosis) was reported in the literature. It developed following long-term transfusions in patients with DBA [15,35]. Some key features of end-stage cardiac hemochromatosis included dilated cardiomyopathy, severe biventricular systolic disfunction, supraventricular and ventricular arrhythmias, as well as cardiogenic shock [15]. 

The pathophysiological hallmark of DBA is defective ribosome synthesis, leading to erythroid aplasia [36]. The DBA’s genetic origin was first established in 1999 when RPS19—a gene that codes for a ribosomal protein—was identified in chromosome 19q13.2 [37]. For almost a decade, *RPS19* seemed to be the only gene involved in the pathogenesis of DBA, but up to the present an additional 27 genes have been identified in relation to DBA [16,38,39]. Although 50–70% of cases of DBA are dominantly inherited, X-linked recessive inheritance patterns have also been reported [3]. Moreover, recent studies strongly support evidence of an autosomal recessive inheritance of DBA [2]. Typical laboratory findings in DBA consist of isolated macrocytic but normochromic anemia, low reticulocyte count, and normocellular marrow, but with a paucity of erythroid precursors [1].

The diagnosis of DBA is based on classical and supporting criteria, which are under constant analysis as newer genetic mutations are identified. Alongside anemia with reticulocytopenia and normocellular bone marrow, onset in the first year of life is also essential for the diagnosis, as more than 90% of cases are reported in this age group [40]. Adult onset is rarely reported and is usually associated with severe transfusion-dependent non-regenerative anemia [41,42].

Several physical abnormalities have been reported in 47% of children with DBA, among which craniofacial anomalies (microcephaly, cleft lip and/or palate, high arched palate, micrognathia, low-set ears, low hairline, epicanthus, ptosis, hypertelorism, congenital glaucoma/cataract, strabismus) and upper limb and hand anomalies (triphalangeal thumb, syndactyly, flat thenar eminence, duplex or bifid thumb) are the most common clinical findings in these patients [43]. Growth retardation occurs in 47% of cases, whereas genitourinary and cardiac anomalies are reported in 30% of cases [1]. According to Da Costa et al., the revealing manifestations of DBA may also be nonspecific and only subsequent to anemia—conjunctival and skin pallor, tachycardia, and a systolic heart murmur [44]. However, Vlachos et al. found a higher prevalence of congenital heart disease in children with DBA than in the general population (14.9% vs. <1%), underlying the importance of genetic counseling [11].

Genetic studies of families with DBA cases show a surprising genotypic diversity. As genetic analysis became more accessible, various genotype–phenotype correlations were reported [45]. In a 128-person unrelated DBA families study, Quarello et al. observed a higher incidence of somatic malformations in children with RPL5 and RPL11 mutations and a significant correlation between RPL5 mutations and craniofacial malformations and between hand malformations and RPL11 mutations [46]. In a 472-person cohort consisting of individuals with confirmed or high suspicion of DBA, Ulirsch et al. observed that 83% of cases with an RPL5 mutation associated with at least one congenital malformation in contrast to RPS19 mutations, which showed any congenital malformation in only 34% of cases [16]. In contrast, Arbiv et al., in a study including 74 patients with DBA, observed that RPL11 mutations showed no increased propensity to develop any malformations, but demonstrated a low incidence of genitourinary malformations in patients with RPS19 mutations [47]. These variations and discrepancies are probably due to small samples and sparse data and do not overshadow the importance of genetic analysis in these patients. Diverse genetic backgrounds and molecular physiology were also reported in children with bradyarrhythmia. Ishikawa et al. reported that several mutations in genes that play important roles in cardiac electrophysiology, heart development and cardioprotection were associated with bradyarrhythmia [48]. Being responsible for cardiac conduction velocity [46,49], calcium regulation [50], sodium channel complex regulation [51], or even cardiac development [52], these genes may be responsible for bradycardia in children without structural abnormalities of the cardiac conduction system, caused by underlying structural heart disease.

Not only should symptomatic children with DBA be considered for further investigations, but also their families need genetic counseling as so-called silent DBA or carriers of RP mutations were found in relatives (sibling or parent) who are not anemic but have isolated macrocytosis, elevated eADA (erythrocyte adenosine deaminase enzyme) levels and/or a genetic mutation of the ribosomal protein [53]. Although asymptomatic, the risk of developing malignant diseases in these silent DBAs emphasizes the importance of follow-up programs [54].

However, genetic analysis is not always available and there are situations when the diagnosis can only be established after ruling out other differential diagnoses. A more common condition than DBA, with acquired etiology (viral or idiopathic) and without corticotherapy indication, is transient erythroblastopenia of childhood. Children with transient erythroblastopenia develop normocytic anemia and are diagnosed usually later than DBA patients (between six months and four years old) [55]. Other conditions which should not be forgotten include viral infections with parvovirus B19, HIV, hepatitis viruses, Epstein–Barr virus, poisoning, drugs, toxins (e.g., sodium valproate, carbamazepine, sulfonamides, isoniazid), and even immune-mediated conditions (e.g., thymomas, systemic lupus erythematosus, myasthenia gravis) [3].

The alterations in ribosome synthesis were associated with a perpetual activation of the TP53 (tumor protein p53) tumor suppressor pathway [2]. Consequently, higher rates of neoplastic disorders were reported in children with DBA. Established 30 years ago, the DBA Registry of North America (DBAR; www.clinicaltrials.gov (accessed on 24 November 2022), #NCT00106015) is the largest international DBA patient cohort and includes 900 participants, providing an essential resource for long-term follow-up in these cases. According to the available data, the most frequent neoplastic conditions include myeloid leukemia, myelodysplastic syndrome, colon cancer, genital cancers, and osteosarcomas as the most frequent conditions. These worrying data emphasize the importance of screening programs for these children. In 2021, Lipton et al. proposed the use of colonoscopy at a five-year interval in children with DBA [56]. The age of the first colonoscopy in these patients is still under debate. However, it has been reported that a premalignant adenomatous polyp was diagnosed in an 8-year-old patient who underwent colposcopy due to unrelated abdominal pain [56].

The management of DBA may be challenging for clinicians: although 80% of the patients respond to the initial corticotherapy, 30% result in transfusion and/or corticosteroid dependency [1]. However, hematopoietic stem cell transplantation is the only curative therapy for DBA. On behalf of the Pediatric Diseases and Severe Aplastic Anemia Working Parties of the European Group for Blood and Marrow Transplantation, Diaz-de-Heredia et al. reinforced that stem cell transplantation should be provided to all children with an available matched family donor who is not a DBA silent phenotype [7]. The available results are encouraging: in 2005, Roy et al. reported, in a study including 41 patients with DBA who underwent bone marrow transplantation from an HLA-identical sibling donor, a three-year probability of overall survival of 64% [57]. Two years later, Mugishima et al. reported a three-year overall survival rate of 100% in children who underwent bone marrow transplantation compared to only 40% in those who underwent umbilical cord blood transplantation [5]. A more recent article published by Miano et al. on behalf of the European Society for Blood and Marrow Transplantation reported that, in a large cohort of 106 patients with DBA, a three-year overall survival and event-free survival rate was 84% [6].

## 6. Conclusions

We present the case of an infant with what could arguably be considered a “diamond” heart, suffering from DBA and severe persistent sinus bradycardia of unknown etiology. While the cardiology prognosis is good, as we consider this a case of persistent but well-tolerated sinus bradycardia, the hematology prognosis remains to be determined by future corticoid response.

## Figures and Tables

**Figure 1 medicina-59-00362-f001:**
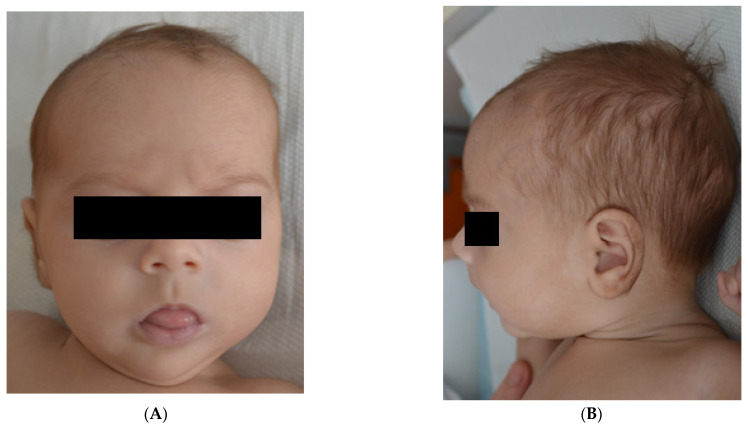
(**A**). Frontal bossing, downslanting palpebral fissures. (**B**). Micrognatism.

**Figure 2 medicina-59-00362-f002:**
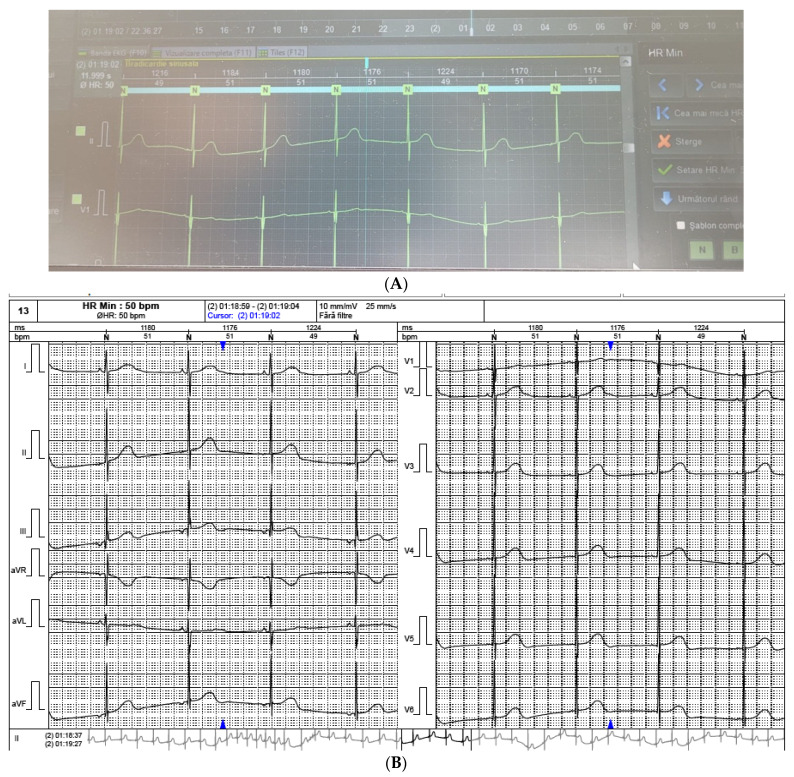
(**A**,**B**). Severe sinus bradycardia during sleep. Minimum heart rate = 49 beats/min.

**Figure 3 medicina-59-00362-f003:**
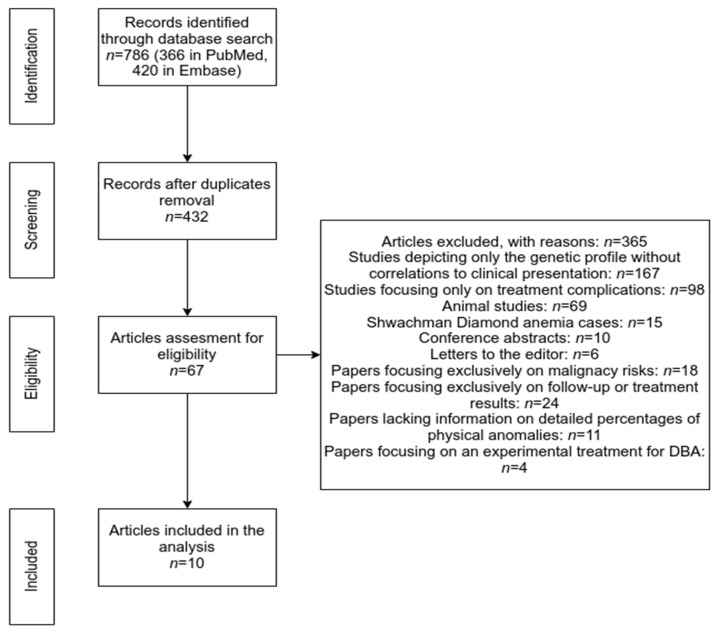
Prisma Flowchart (methodology).

**Table 1 medicina-59-00362-t001:** Phenotypic and genotypic profile in DBA reported cases.

Publication (Author, Year)	Country	Sample Size	Phenotypic Characteristics	Genotypic Characteristics
Ulirsch et al., 2018 [16]	Various: USA (176), Poland (67), Turkey (16)	472	44.4% with physical malformations: 18.5% craniofacial, 15.4% limbs, 6.6% genitourinary, 14.3% cardiac, 8.7% short stature	RPS19 mutations (58%); congenital malformations more commonly associated with RPL5 (83%) or RPL11 (73%) than RPS19 (34%)
Quarello et al., 2020 [17]	Italy	283	50.5% with physical malformations: 17% craniofacial, 16% upper limbs anomalies, 15% cardiac defects	RPS19 mutations (30%), RPL5 mutations (12.5%), RPL11 (9.5%) 26/271), RPS26 (8.5%), RPL35A (3%), RPS17 (3%), and RPS24 (1%); craniofacial and upper limb malformations were more frequent in RPL genes mutations
Pavlova et al., 2020 [18]	Russia	187	40.2% with physical malformations: craniofacial dimorphism, heart malformations, short stature, short neck	The most common mutations: RPS19, RPL5, and RPS7; cleft palate and/or cleft lip anomalies were associated with *RPL5* gene mutations (4/6 cases)
Wan et al., 2021 [19]	Japan	104	19.6% with physical malformations: 31.6% finger anomalies, 31.6% cardiovascular and urogenital system	RP mutations (58.3%): 31.2 % RPS19, 8.3% RPS26, 6.3% RPL5, 4.2% RPS24, 4.2% RPL11, 2.1% RPS7 and RPL35a
Fatima et al., 2020 [20]	Pakistan	74	34% with physical malformations: 23.5% cleft lip and palate, 20.5% thumb and upper limb defects, 14.7% craniofacial anomalies, 8.8% cardiac, 5.8% urogenital malformations	Genetic studies not conducted
Volejnikova et al., 2020 [21]	Slovak Republic and Czech Republic	62	66% with physical malformations: thumb anomalies and/or thenar hypoplasia (32%), craniofacial dimorphism (31%) and congenital heart defects (23%)	RPS19 mutations (31%), RPL5 (14.5%), RPL11 (14.5%), RPS26 (8%), deletions involving some of the RP genes (6%), point mutations in RPS 7 (5%) and RPS17 (2%); tumors/MDS occurred exclusively in patients with RPL11, RPL5 or RPS19 mutations
Giri et al., 2018 [22]	United States	40	33% with physical malformations: microcephaly (28%), short stature (33%), cardiac defects (28%) and/or urogenital abnormalities (23%)	RPL35A mutations (52.5%), single-nucleotid variants or small indels (47.5%); learning difficulties, craniofacial abnormalities, skeletal and limb defects more frequent in patients with large deletions
Vogel et al., 2021 [23]	Switzerland	17	77% with physical malformations: 17% lower limb, 12% anorectal	RPL mutation: more frequently associated with physical malformations and milder anemia compared to RPS mutation
Mininni et al., 2022 [24]	Argentina	15	80% with physical malformations: 66% short stature, 40% head and neck, 20% thumb, 13% cardiac	RPS19 mutations (30%), RPS26 mutations (20%), other RP genes (40%); no therapy required in GATA1 and RPL5 (2 cases)
Delaporta et al., 2014 [25]	Greece	7	71% with physical malformations	RPS19 mutations (35.2%); all 3 patients with *RPL5* gene mutations had congenital abnormalities, in contrast with only 1/3 patients with RPS19 mutations

## Data Availability

The data presented in this study are available on request from the corresponding author.

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
