# Peer review of "Unusual Association of Diamond–Blackfan Anemia and Severe Sinus Bradycardia in a Six-Month-Old White Infant: A Case Report and Literature Review"

_medicina, 2023, doi:10.3390/medicina59020362_

Round 1

Reviewer 1 Report

Minor corrections:

1) Page 1, line 40:  correct "tollerated" to "tolerated"

2) Page 2, line 59: DBA is characterised "with" a broad spectrum.. Write "by" in place of "with" 

3) Page 12, line 287: "complexregulation" correct as "complex regulation"

4) Page 12, line 327:  correct "tollerated" to "tolerated"

These are to support my recommendation of minor:

What is the main question addressed by the research? - Brady cardia can co-exist with DBS

Is it relevant and interesting? -  Yes

How original is the topic? -  Yes

What does it add to the subject area compared with other published material? -  Association of bradycardia with DBS

Is the paper well written? - Yes

Is the text clear and easy to read? - yes

Are the conclusions consistent with the evidence and arguments presented? - yes

Do they address the main question posed? - yes

Author Response

We would like to thank the Reviewer and the Editorial board for all the positive remarks regarding our work. We are delighted to hear that the Editor / Reviewer observed the quality of the manuscript and the in-depth analysis of the subject.

We assure the EIC that we have read carefully every suggestion from this decision letter and tried our best to improve the quality of the document accordingly.

Q1: Page 1, line 40:  correct "tollerated" to "tolerated"

Answer:

Thank you for your remark!

We modified the text accordingly:

“We consider this a case of well tolerated sinus bradycardia and foresee a good cardiologic prognosis, while the hematologic prognosis remains determined by future corticoid response”.

Q2: Page 2, line 59: DBA is characterised "with" a broad spectrum.. Write "by" in place of "with"

Answer:

Thank you for your excellent suggestion!

We modified the text according to the Reviewer suggestion, as follows:

“DBA is characterized by a broad spectrum of phenotypes”.

Q3: Page 12, line 287: "complexregulation" correct as "complex regulation"

Answer:

Thank you for your observation!

We updated the manuscript accordingly:

“Being responsible for cardiac conduction velocity [36,39], calcium regulation [40], sodium channel complex regulation”.

Q4: Page 12, line 327:  correct "tollerated" to "tolerated"

Answer:

Thank you for your excellent remark!

Following the Reviewer remark, we modified the main text accordingly:

“While the cardiology prognosis is good, as we consider this a case of persistent but well tolerated sinus bradycardia”.

Reviewer 2 Report

Review of the publication “Unusual association of Diamond–Blackfan anemia and severe sinus bradycardia in a six-month-old Caucasian infant: a case report and literature review” by Moisă et al.

Thank you for the opportunity to review the aforementioned publication.

This case report presents an unusual association of Diamond-Blackfan anemia (DBA) with severe sinus bradycardia in a 6-month-old infant.

There is a major concern with the publication:

Line 33 – the Authors suggest that the bradycardia was spontaneous, however it occurred during CVC insertion: “The bradycardia appeared after mild sedation for central venous line insertion but did not resolve once the sedative was eliminated from the body” (line 211, 212). This one sentence says more about possible cause of severe bradycardia than the entire differential diagnosis that was performed. Bradycardia or even a sinus arrest is listed as possible complications of CVC insertion. The association between DBA and sinus bradycardia that the Authors claim in fact is very likely to be a known complication of CVC insertion. Taking into account temporal relationship between CVC insertion and the moment when bradycardia appeared, complication of CVC insertion in my opinion is the most likely cause, unlike trying to associate DBA with bradycardia in this infant. The infant was not bradycardic before CVC insertion. I am surprised that 14 authors of the publication did not get noticed this.

Here is some literature on the topic of bradycardia due CVC insertion:

1.       Saraschandra Vallabhajosyula , Bernardo Selim. Sustained Bradycardia Following Internal Jugular Catheter Insertion. American Journal of Respiratory and Critical Care Medicine 2016;193:A1981,

2.       Gapp J, Krishnan M, Ratnaraj F, Schroell RP, Moore D. Cardiac Arrhythmias Resulting from a Peripherally Inserted Central Catheter: Two Cases and a Review of the Literature. Cureus. 2017 Jun 3;9(6):e1308. doi: 10.7759/cureus.1308. PMID: 28690942; PMCID: PMC5497924,

3.       Nazinitsky A, Covington M, Littmann L. Sinus arrest and asystole caused by a peripherally inserted central catheter. Ann Noninvasive Electrocardiol. 2014 Jul;19(4):391-4. doi: 10.1111/anec.12116. Epub 2013 Nov 29. PMID: 24286255; PMCID: PMC6932029.

If the fact that severe bradycardia developed during CVC insertion was not neglected, perhaps the genetic testing for cardiac arrhythmia would not be necessary.

Other concerns:

Line 26/lone 47 – the Authors mention that DBA is a rare condition. Whenever rare conditions are reported there should be estimation of their incidence given, e.g. number of new cases per million newborns. The information that condition is rare in my opinion is not sufficient.

Line 26 – the Authors state that DBA is “usually inherited condition”. From what I know DBA as a type if inherited aplastic anemia. Are there cases of acquired DBA in literature?

Line 30 – was the reason for admission at the age of 6 month more infection than anemia?  Obviously anemia could lead to infection, but the primary diagnosis was probably infection.

Line 32 – do we know what was the anatomical site of infection – was this gastrointestinal infection? If yes the expression diarrhea is not correct here. 

Line 33/line 91 – the unit for Hb concentration is incorrect, there should be 3 g/dL, even better 30 g/L.

Line 42 – the hematological prognosis in this infant is not only determined by the response to corticosteroids. If corticosteroids are not effective there are other methods of treatment, including auto hematopoietic cell transplantation.

The fact that the authors did not take into account complication of CVC insertion as a possible cause of severe bradycardia in this infant makes all literature search and conclusions irrelevant. Moreover there are other concerns, some of which I pointed out above.

Author Response

Review of the publication “Unusual association of Diamond–Blackfan anemia and severe sinus bradycardia in a six-month-old Caucasian infant: a case report and literature review” by Moisă et al.

Thank you for the opportunity to review the aforementioned publication.

This case report presents an unusual association of Diamond-Blackfan anemia (DBA) with severe sinus bradycardia in a 6-month-old infant.

Answer:

We would like to thank the Reviewer and the Editorial board for all the positive remarks regarding our work. We are delighted to hear that the Editor / Reviewer observed the quality of the manuscript and the in-depth analysis of the subject.

We assure the EIC that we have read carefully every suggestion from this decision letter and tried our best to improve the quality of the document accordingly.

Specific comments:

There is a major concern with the publication:

Q1: Line 33 – the Authors suggest that the bradycardia was spontaneous, however it occurred during CVC insertion: “The bradycardia appeared after mild sedation for central venous line insertion but did not resolve once the sedative was eliminated from the body” (line 211, 212). This one sentence says more about possible cause of severe bradycardia than the entire differential diagnosis that was performed. Bradycardia or even a sinus arrest is listed as possible complications of CVC insertion. The association between DBA and sinus bradycardia that the Authors claim in fact is very likely to be a known complication of CVC insertion. Taking into account temporal relationship between CVC insertion and the moment when bradycardia appeared, complication of CVC insertion in my opinion is the most likely cause, unlike trying to associate DBA with bradycardia in this infant. The infant was not bradycardic before CVC insertion. I am surprised that 14 authors of the publication did not get noticed this.

Here is some literature on the topic of bradycardia due CVC insertion:

  1. Saraschandra Vallabhajosyula , Bernardo Selim. Sustained Bradycardia Following Internal Jugular Catheter Insertion. American Journal of Respiratory and Critical Care Medicine 2016;193:A1981,

  1. Gapp J, Krishnan M, Ratnaraj F, Schroell RP, Moore D. Cardiac Arrhythmias Resulting from a Peripherally Inserted Central Catheter: Two Cases and a Review of the Literature. Cureus. 2017 Jun 3;9(6):e1308. doi: 10.7759/cureus.1308. PMID: 28690942; PMCID: PMC5497924,

  1. Nazinitsky A, Covington M, Littmann L. Sinus arrest and asystole caused by a peripherally inserted central catheter. Ann Noninvasive Electrocardiol. 2014 Jul;19(4):391-4. doi: 10.1111/anec.12116. Epub 2013 Nov 29. PMID: 24286255; PMCID: PMC6932029.

If the fact that severe bradycardia developed during CVC insertion was not neglected, perhaps the genetic testing for cardiac arrhythmia would not be necessary.

Answer:

Thank you for your excellent point!

The Reviewer is right that central venous catheter (CVC) insertion might cause cardiac arrhythmias. They are primarily driven by guidewire or CVC direct mechanical irritation. Nevertheless, atrial premature contractions, atrial tachycardia, atrial fibrillation, ventricular tachycardia and ventricular fibrillation were the most common arrhythmias reported in the literature. Also, there were reported only sporadic cases of bradyarrhythmias due to CVC insertion, including in adult patients (reference mentioned by the Reviewer: Saraschandra Vallabhajosyula, Bernardo Selim. Sustained Bradycardia Following Internal Jugular Catheter Insertion. American Journal of Respiratory and Critical Care Medicine 2016;193:A1981).

However, CVC insertion – linked cardiac arrhythmias has a much lower incidence in children than in adults. In a large cohort of pediatric patients (3180 inserted catheters), all types of cardiac arrhythmias occurred in only 31 children (1% cases): atrial arrhythmias (71%), ventricular arrhythmias (13%) and undetermined (16%) (DOI: 10.1007/s00246-019-02274-1). Thus, bradyarrhythmias are expected to occur in much lower than 1% of cases (atrial arrhythmias being the most common).

There were published limited data on bradycardia following CVC insertion, mainly case reports in adult patients (references mentioned by the Reviewer: DOI: 10.7759/cureus.1308 and DOI: 10.1111/anec.12116). Therefore, establishing a direct correlation between CVC insertion in pediatric patients and bradyarrhythmias might be misleading due to the lack of data in this subset of patients.

Moreover, bradycardia in our case report occurred immediately after mild sedation (before CVC insertion). Also, the bradycardia persisted after CVC extraction, which makes improbable the association between cardiac arrhythmia and CVC insertion. Thus, we referred to bradycardia as of uncertain origin in the discussion section, as it is impossible to link the arrhythmia to one causal factor. In addition, the genetic testing revealed the presence of a heterozygous mutation of the DSG2 gene, which is associated with autosomal dominant arrhythmogenic right ventricular cardiomyopathy (ARVC). ARVC could cause bradyarrhythmias in adult patients, as was documented in one study (DOI: 10.1016/j.amjcard.2019.02.032), but it could be misleading to link directly ARVC to bradycardia in young children. Also, until new information appears in the literature related to DSG2 gene mutations involved in the occurrence of heart rhythm disorders, we cannot state with certainty that the gene mutation detected in our patient's case is correlated with the appearance of bradycardia.

Nevertheless, the discussion on bradycardia origin is useful and interesting, and could provide directions for further research (especially involving these rare conditions). Therefore, we updated the manuscript as follows:

“One study involving a large cohort of pediatric patients with central venous cath-eters (CVC) (n = 3180), reported a low incidence of all types of cardiac arrhythmias (1% of cases): atrial arrhythmias (71%), ventricular arrhythmias (13%) and undetermined (16%) [25]. Thus, the incidence of bradycardia following CVC insertion is expected to be much lower than 1%. Also, there were published limited data on bradycardia following CVC insertion, mainly case reports in adult patients [26,27]. Therefore, establishing a direct correlation between CVC insertion and bradyarrhythmias in pediatric patients might be misleading due to the lack of data in this subset of patients.”

Other concerns:

Q2: Line 26/lone 47 – the Authors mention that DBA is a rare condition. Whenever rare conditions are reported there should be estimation of their incidence given, e.g. number of new cases per million newborns. The information that condition is rare in my opinion is not sufficient.

Answer:

Thank you for the raised point!

We agree with the Reviewer that the incidence of DBA should be reported in order to provide a support for the affirmation. Consequently, we modified the manuscript according to the Reviewer recommendations:

“Diamond–Blackfan anemia is a rare (6-7/million live births)”.

Q3: Line 26 – the Authors state that DBA is “usually inherited condition”. From what I know DBA as a type if inherited aplastic anemia. Are there cases of acquired DBA in literature?

Answer:

Thank you for your excellent remark!

The Reviewer is right that DBA is a type of inherited aplastic anemia. As far as we know, there were no reported cases of acquired DBA in the literature. Consequently, the removed the wors “usually” as it could be misleading, as follows:

“Diamond–Blackfan anemia is a rare (6-7/million live births), inherited condition manifesting as severe anemia due to impaired bone marrow production of red blood cells.”

Q4: Line 30 – was the reason for admission at the age of 6 month more infection than anemia?  Obviously anemia could lead to infection, but the primary diagnosis was probably infection.

Answer:

Thank you for your suggestion!

The Reviewer is right regarding the importance of infection to patient admission. However, at hemoglobin concentration of 3 g/dL it is hard to point out a single causal factor for hospital admission, as severe anemia and infection were overlapping. Therefore, we reported mixed causes for patient admission:

“She presents with severe anemia causing hypoxia which in turn caused severe dyspnea and polypnea, that agreeably had mixed causes (hypoxic and infectious), as the child was febrile”.

Also, we highlighted the importance of infection, as a reason for admission in the main text, as follows:

“The current presentation was motivated by a fever (38 degrees Celsius) that started 3 days before being admitted and was accompanied by feeding refusal and accentuated paleness.”

Q5: Line 32 – do we know what was the anatomical site of infection – was this gastrointestinal infection? If yes the expression diarrhea is not correct here.

Answer:

Thank you for pointing this out!

We agree with the Reviewer that the term diarrhea would be inappropriate if gastrointestinal tract was the anatomical site of the infection. However, diarrhea could have different causal factors in young children, including fever. Also, diarrhea developed late after fever onset, and we could not link for certain diarrhea to a gastrointestinal tract infection. Therefore, we used the term diarrhea as a multifactorial symptom, instead of gastrointestinal tract infection.

Q6: Line 33/line 91 – the unit for Hb concentration is incorrect, there should be 3 g/dL, even better 30 g/L.

Answer:

Thank you for your excellent observation!

The Reviewer is right and we corrected the unit for hemoglobin concentration in the above-mentioned lines:

“Severe anemia (hemoglobin <3 g/dL)”

“Initial biological tests revealed severe anemia (hemoglobin (Hb) level of less than 3 g/dL)”.

Q7: Line 42 – the hematological prognosis in this infant is not only determined by the response to corticosteroids. If corticosteroids are not effective there are other methods of treatment, including auto hematopoietic cell transplantation.

Answer:

Thank you for your great point!

We agree with the Reviewer that hematologic prognosis is not limited to corticosteroid therapy response. Clinical studies reported other factors which could negatively impact hematologic prognosis, such as treatment related complications (the most common – hemosiderosis) or risk of leukemia, which is profoundly increased in DBA patients (a 200 higher relative risk) (DOI: 10.1097/00005792-199603000-00004). Consequently, we modified the abstract in the line with the Reviewer suggestion:

“We consider this a case of well tolerated sinus bradycardia and foresee a good cardiologic prognosis, while the hematologic prognosis remains determined by future corticoid response, treatment-related complications and risk of leukemia”.

Q8: The fact that the authors did not take into account complication of CVC insertion as a possible cause of severe bradycardia in this infant makes all literature search and conclusions irrelevant. Moreover there are other concerns, some of which I pointed out above.

Answer:

Thank you for your excellent remarks!

We agree with the Reviewer that CVC insertion as a potential cause of bradycardia should have been discussed in the manuscript. As we mentioned above, it could be misleading to point out CVC insertion as a single determinant factor of bradycardia in the reported clinical case due to several factors:

- only 1% of pediatric patients with CVC had cardiac arrhythmias, as reported in clinical studies (the most common was atrial arrhythmias followed by ventricular arrhythmias – thus, incidence of bradyarrhythmias is much lower than 1%);

- there were reported limited data on bradycardia followed by CVC insertion in adults (mainly case reports). Extrapolation of these data to pediatric population could be inappropriate;

- bradycardia in our case report occurred immediately after mild sedation (before CVC insertion);

- bradycardia persisted after CVC extraction, which makes improbable the association between cardiac arrhythmia and CVC insertion.

We also updated the discussion section with new data regarding the potential link between CVC insertion and bradycardia, thus addressing some issues which were not currently investigated in the literature:

“One study involving a large cohort of pediatric patients with central venous cath-eters (CVC) (n = 3180), reported a low incidence of all types of cardiac arrhythmias (1% of cases): atrial arrhythmias (71%), ventricular arrhythmias (13%) and undetermined (16%) [25]. Thus, the incidence of bradycardia following CVC insertion is expected to be much lower than 1%. Also, there were published limited data on bradycardia following CVC insertion, mainly case reports in adult patients [26,27]. Therefore, establishing a direct correlation between CVC insertion and bradyarrhythmias in pediatric patients might be misleading due to the lack of data in this subset of patients”.

Reviewer 3 Report

this is a case report. well-structured and well-described case

Author Response

this is a case report. well-structured and well-described case

Answer:

We would like to thank the Reviewer and the Editorial board for all the positive remarks regarding our work. We are delighted to hear that the Editor / Reviewer observed the quality of the manuscript and the in-depth analysis of the subject.

We assure the EIC that we have read carefully every suggestion from this decision letter and tried our best to improve the quality of the document accordingly.

Reviewer 4 Report

The authors present case report of unusual association of DiamondBlackfan anemia and severe sinus bradycardia in a six-month-old infant with literature review. Diamon-Blackfan anemia was well described before and many studies of large groups of patients has been published before. However, in this manuscript the authors described unusual association of DiamondBlackfan anemia and severe sinus bradycardia. The case is well described. My main concern is lack of the information about the hematologic response to steroids. Is the patients transfusion dependent? If yes, how often does he need transfusion? How long is the follow-up?

The literature is deeply reviewed. The discussion is well written.

Author Response

The authors present case report of unusual association of Diamond–Blackfan anemia and severe sinus bradycardia in a six-month-old infant with literature review. Diamon-Blackfan anemia was well described before and many studies of large groups of patients has been published before. However, in this manuscript the authors described unusual association of Diamond–Blackfan anemia and severe sinus bradycardia. The case is well described.

Answer:

We would like to thank the Reviewer and the Editorial board for all the positive remarks regarding our work. We are delighted to hear that the Editor / Reviewer observed the quality of the manuscript and the in-depth analysis of the subject.

We assure the EIC that we have read carefully every suggestion from this decision letter and tried our best to improve the quality of the document accordingly.

Q1: My main concern is lack of the information about the hematologic response to steroids. Is the patients transfusion dependent? If yes, how often does he need transfusion? How long is the follow-up?

Answer:

Thank you for your excellent questions!

Patient was discharged (not transfusion-dependent) with recommendation of corticosteroid therapy - prednisone 2 mg/kg/day, in association with proton pump inhibitor for three weeks. Afterwards, the hematologic response, as well as requirement of long-term corticosteroids or transfusions will be evaluated. Although 80% respond to the initial corticosteroid therapy, 30% of cases require long-term transfusion or corticosteroid therapy. These patients should followed-up lifelong, as hematopoietic stem cell transplantation is the only curative treatment for DBA. Moreover, many patients could relapse after several years of evolution and might became transfusion-dependent (https://www.jacc.org/doi/10.1016/S0735-1097%2822%2903261-2), with worse prognosis due to iron overload (including end-stage cardiac hemochromatosis). We updated the manuscript accordingly:

“At discharge, patient was not transfusion-dependent, with recommendation of corticosteroid therapy - prednisone 2 mg/kg/day, in association with proton pump inhibitor for three weeks. Afterwards, hematologic response, as well as the requirement of long-term corticosteroids or transfusions will be evaluated. She will be followed-up lifelong, as many patients could relapse after several years of evolution and might became transfusion-dependent, with worse prognosis due to iron overload (including end-stage cardiac disease) [15].”

Q2: The literature is deeply reviewed. The discussion is well written.

Answer:

We want to thank the Reviewer for observing the quality of the manuscript and the literature review. Our case report and literature review contain valuable data for DBA patients, including a previously unreported clinical association – sinus bradycardia. Also, we believe that updating the paper in the line with the Reviewer recommendations and suggestions improved its overall quality and the reading process.

Round 2

Reviewer 2 Report

Although my initial suggestion was to reject the publication, the Authors somewhat resolved my concerns on the links between CVC insertion and bradycardia. The fact that "bradycardia in our case report occurred immediately after mild sedation (before CVC insertion" should be provided in the abstract as well as case discription and discussion. It is of utmost importance here to make the link between CVC insertion and bradycardia as weak as possible, especially that the Authors report unusual symptom in a infant with a very rare condition. The fact that bradycardia persisted following CVC removal could have been caused by the damage to the sinus node by the guidewire, but if the bradycardia occured before CVC insertion makes it less likely. I would like the Authors to provide the readers with the  information on HR at the following time points: before sedation, after administration of sedative, directly before guidewire insertion, after CVC placement, following sedation. I would like also to see the description of the the procedure itself: technique used, vein catheterized, catheter used, sedative agent used (including dose per kg). These information should be included in the case description. These information are important to make the link between CVC insertion and bradycardia as weak as possible, as I mentioned earlier. My other suggestions have been implemented by the Authors.

Thank you.

Author Response

Although my initial suggestion was to reject the publication, the Authors somewhat resolved my concerns on the links between CVC insertion and bradycardia.

Answer:

We want to thank the Reviewer for observing the quality of the manuscript and the literature review. Our case report and literature review contain valuable data for DBA patients, including a previously unreported clinical association – sinus bradycardia. Also, we believe that updating the paper in the line with the Reviewer recommendations and suggestions improved its overall quality and the reading process.

The fact that "bradycardia in our case report occurred immediately after mild sedation (before CVC insertion" should be provided in the abstract as well as case discription and discussion. It is of utmost importance here to make the link between CVC insertion and bradycardia as weak as possible, especially that the Authors report unusual symptom in a infant with a very rare condition. The fact that bradycardia persisted following CVC removal could have been caused by the damage to the sinus node by the guidewire, but if the bradycardia occured before CVC insertion makes it less likely.

Answer:

Thank you for the point raised!

The Reviewer is right that the information concerning bradycardia timing in relation to CVC insertion should be also provided in the abstract, case description and discussion sections. Consequently, we modified the manuscript accordingly:

  1. a) abstract: “she developed severe persistent sinus bradycardia immediately after mild sedation (before central venous catheter insertion), not attributable to any of the more frequent causes, with heart rates as low as 49 beats/minute on 24-hour Holter monitoring, less than the first percentile for age, but with a regular QT interval and no arrhythmia”.
  2. b) case description: “The patient was initially tachycardic, but she became bradycardic immediately after mild sedation for central line insertion (before central venous catheter (CVC) insertion)”.
  3. c) discussion section: “Moreover, the bradycardia in our case occurred before CVC insertion”.

I would like the Authors to provide the readers with the  information on HR at the following time points: before sedation, after administration of sedative, directly before guidewire insertion, after CVC placement, following sedation.

Answer:

Thank you for your great suggestion!

We agree with the Reviewer regarding the necessity of providing additional information on heart rate at specific time points in relation to CVC insertion. Therefore, we updated the manuscript with available data on heart rate at different time frames, as follows:

“Concerning different time points, the following heart rates were recorded: 80/minute before sedation (systolic blood pressure 80-90 mmHg), 47/minute before CVC insertion and 50-80/minute after CVC insertion and in the days afterwards (systolic blood pressure 85-95 mmHg)”.

I would like also to see the description of the the procedure itself: technique used, vein catheterized, catheter used, sedative agent used (including dose per kg). These information should be included in the case description. These information are important to make the link between CVC insertion and bradycardia as weak as possible, as I mentioned earlier.

Answer:

Thank you for your excellent point!

We agree with the Reviewer that supplementary information regarding the CVC insertion procedure itself should be provided. In consequence, we updated the manuscript (case presentation section) with additional available data on the procedure, as follows:

“A central venous catheter FR 18 was inserted using the following sedation strategy: ketamine 15 mg (2.5 mg/kg), midazolam 0.75 mg (0.125 mg/kg) and atropine 0.05 mg”.

My other suggestions have been implemented by the Authors.

Thank you very much!